# Reducing Manual Workload in SAR-Based Oil Spill Detection Through Uncertainty-Aware Deep Learning

Dina S. Solskinnsbakk[1,2], Sigurd A. Hanssen[1,2], Harald L. Joakimsen[1], Vilde B. Gjærum[2], Elisabeth Wetzer[1], and Kristoffer K. Wickstrøm*[1]

[1]UiT The Arctic University of Norway
[2]Kongsberg Satellite Services

## Abstract

Constant monitoring of the oceans is required to detect oil spills and reduce environmental damage associated with spills. Synthetic Aperture Radar (SAR) imaging is a critical tool for oil spill detection, but is complex and requires time-consuming manual labor for analysis. Deep learning has shown encouraging performance in automatic classification of oil spills on these images, but the performance is not yet sufficient for a deep learning classifier to act autonomously, making manual assessment essential. However, if only a reduced subset of uncertain samples had to be analyzed by human experts while the remaining samples could be automatically classified, it could greatly reduce the manual workload. In this study, we investigate if uncertainty estimates can identify which samples should be prioritized for manual inspection. Specifically, we propose a novel pipeline of defining a user-specified error tolerance and identifying an uncertainty threshold that filters out samples for automatic/manual thresholding. We evaluate the proposed pipeline on challenging real-world data. The results show that our proposed uncertainty-based ranking technique can reduce the manual workload by 41%, paving the way for new and more efficient ways to detect marine oil spills.

## 1 Introduction

Marine oil spills are common, with several thousand spills occurring each year in the United States [1]. It can have major environmental impact due to the damage on the marine ecosystem and the wildlife at both the sea and shore [1]. Therefore, the ocean needs to be monitored to minimize the damage by removing them quickly after release. There are multiple ways of large-scale monitoring of the ocean, but Synthetic Aperture Radar (SAR) imaging is often a preferred choice [2]. SAR is independent of daylight and cloud coverage while covering large areas, which is a big advantage compared to other options such as optical imagery [2].

Despite all the advantages of SAR imaging, the resulting images are complex and cover large areas,

---

*Corresponding Author: kwi030@uit.no

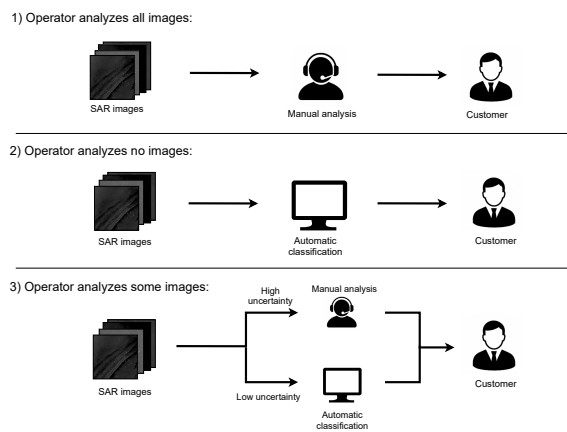

**Figure 1.** Three scenarios for SAR-based oil spill detection: (1) human analysis with low error rate but high workload, (2) automatic analysis with high error rate but low workload, and (3) combined human and machine analysis for low error and low workload. Images in figure are artificially enhanced SAR images. © Copernicus Sentinel data, processed by KSAT.

making the manual image analysis time-consuming. Automatic systems based on deep learning models have shown encouraging performance for this specific application [2, 3] and could potentially reduce the manual workload. Even though the DL-based systems achieve high accuracy [3–5], it is challenging to put them into operational use-cases because of the harsh consequences of a false negative. However, the vast majority of images are still correctly classified. Therefore, if there was a process that could reliably identify samples that required human evaluation and let the remaining samples be automatically classified, it could significantly reduce the manual workload. The bottom row of Figure 1 illustrates how such a pipeline could be constructed.

In this work, we propose an uncertainty-guided selection process for identifying SAR images of the ocean with potential oil spills detected by a deep learning model that must be processed by human evaluators to ensure sufficient performance. It is well-known that relying on the softmax output of deep learning models is ill-advised due to their high degree of overconfidence [6], and that more sophisticated uncertainty estimation techniques can more

Proceedings of the 7th Northern Lights Deep Learning Conference (NLDL), PMLR 307, 2026.

accurately identify samples that are likely to be misclassified [7]. The key idea in our proposed selection process is to automatically identify an uncertainty threshold that ensures a certain performance, where all samples above the threshold are sent for human evaluation. This approach allows for utilizing all the benefits of DL models without having to achieve perfect performance, while minimizing the risk of false negatives. Additionally, the automated process has the added benefit of minimizing inconsistent analysis due to human error in simple cases. Our contributions are:

- A new procedure in automated deep learning-based classification for uncertainty-based selection of samples for human evaluation.

- Identification of domain-relevant augmentation strategies to allow for uncertainty estimation at test-time.

- An in-depth analysis of the proposed pipeline on real-world challenging data. Our results show that using uncertainty-based selection for human evaluation can significantly reduce the manual workload of operators.

## 2 Related Work

Automatic detection of oil spills from SAR images has been extensively studied [2]. These approaches often revolve around sophisticated feature extraction techniques in combination with classical classification algorithms [8, 9]. More recently, such methods have been outperformed by deep learning-based approaches [2]. Bianchi et al. [3] proposed a deep learning model based on convolutional neural networks with encouraging performance. More recently, Trujillo-Acatitla, Tuxpan-Vargas, et al. [10] also demonstrated the high potential for deep learning-based oil spill detection but across a wider range of deep learning architectures. While all of these works demonstrate that automatic systems have great potential for alleviating the manual workload associated with oil spill detection, we are not aware of any works that have considered how uncertainty estimation could be practically integrated into the oil spill detection pipeline. However, uncertainty modeling has been used extensively in a related manner to improve decision making and human-AI collaboration [11, 12].

## 3 Reducing Manual Workload With Uncertainty-Filtering

Here, we present our proposed approach for filtering out samples that requires human evaluation.

### 3.1 Uncertainty-Filtering With Test-Time Augmentation

In this work, we focus on test-time augmentation (TTA) [13] to model uncertainty in the prediction $\hat{y}$ of a deep learning model $f$. This choice is motivated by the flexibility of TTA, as it requires no modifications to the model like Monte Carlo Dropout [14] or SWAG [15], nor does it require storing multiple models like ensemble approaches [16]. This is highly beneficial in industrial applications, where a working model might already be in place and it is undesirable to alter the existing pipeline solely for the uncertainty estimation. TTA works by generating augmented views of an input and aggregating predictions across all augmented views, and has demonstrated impressive performance across a wide range of applications [13, 17]. This can be mathematically described as taking an input $\mathbf{x}$ and transforming it using a stochastic augmentation procedure $\mathcal{T}$ that produces augmented versions $\tilde{\mathbf{x}}$. Assuming $M$ augmentations are generated, a set of $M$ predictions $\{\hat{y}_1, \cdots, \hat{y}_M\}$ are made. The uncertainty associated with $f$'s prediction of $\mathbf{x}$ is calculated as:

$$\sigma^{(tta)} = \sqrt{\frac{1}{M-1} \sum_{m=1}^{M} (\hat{y}_m - \bar{y})^2}, \qquad (1)$$

where $\bar{y}$ is the mean of the $M$ predictions.

**Choosing augmentations in TTA** A key aspect of TTA is choosing a suitable data augmentation procedure that fits the data and task at hand. A common choice is to apply dropout [18] on the input to generate augmented samples [13], which is motivated by its flexibility, speed, and good performance in many cases [13]. However, in the case of oil spill detection from SAR images, we hypothesize that dropout is not the most effective choice for augmentation. Oil spills typically appear as dark areas in SAR images, and dropping these pixels would do little to alter their appearance, which would therefore induce less informative uncertainty estimates. Also, a high dropout rate could create areas that appear similar to oil spills. Therefore, we instead propose to use augmentations that are more suitable for the particular task and data at hand.

The pixel-value shift (PVS) method shifts each pixel based on the average pixel intensity in the training set. PVS has demonstrated encouraging results for hyperspectral imaging [19], which is often attributed to its ability of generating in-distribution samples. Mathematically, PVS generates samples by

$$\mathbf{x}^{(pvs)} = \mathbf{x} \pm \gamma \cdot \boldsymbol{\mu}^{(tr)}, \qquad (2)$$

where $\boldsymbol{\mu}^{(tr)}$ is the average pixel intensity estimated across the training set and $\gamma$ is a coefficient that

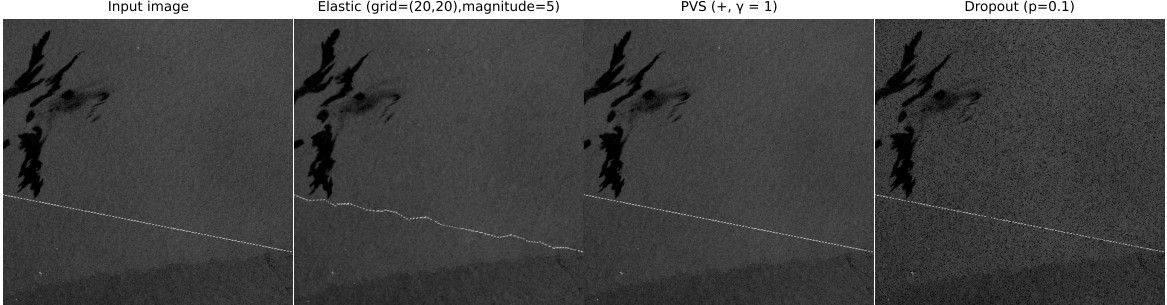

**Figure 2.** An illustration of three types of data augmentation when applied to a SAR image with oil spills. © Copernicus Sentinel data, processed by KSAT

controls the strength of the shift. This coefficient is randomly sampled for each generated sample to induce different shifts for the same sample, and the sampling procedure for $\gamma$ is an important hyperparameters for this type of augmentation.

Elastic transformations are transformations that alter the geometry of an image, which is accomplished by generating displacement vectors for all pixels based on random offsets. The displacement vectors are added to the image through an identity grid that performs the transformation. Our choice of the elastic transformation in the context of oil spills is motivated by the fact that oil spills often exhibit a curved shape, and the elastic transformation generates variation of such curved shapes in a realistic looking manner.

Figure 2 shows each of the augmentations applied to a SAR image containing oil spills. The elastic transformation slightly distorts the content of the image, which is apparent both on the oil spills in the top left corner and for the white line (interference from ground radar) that crosses the image. The difference induced by PVS is more difficult to see, since the change is on the pixel values. As for dropout, the image appears more noisy with more dark spots across the entire image.

## 3.2   Uncertainty-Sorting Procedure

Our core idea is to send all samples that exceed a threshold of uncertainty to manual analysis, assuming that the human analysis will result in only correct predictions. The threshold can be adjusted in such a way that the overall system can obtain a desirable trade-off between the error rate and the reduction in manual workload. First, we assume an independent validation set $X^{(val)} = \{\mathbf{x}_i\}_{i=1}^{N_{val}}$ with $N_{val}$ samples, and that this validation set is sorted from least to most uncertain according to a corresponding set of uncertainty estimates $U^{(val)} = \{\sigma_i\}_{i=1}^{N_{val}}$.

Next, we assume a function that measures the error rate of the classifier for a given set of samples, in this case Er $= 1 -$ accuracy. As part of the procedure, a user specified error rate $e$ must be

provided that indicates an acceptable error for the classifier $f$. The complete procedure is described in Listing 1, and the output of the algorithm is the uncertainty threshold $\tau_e$ that can be used to sort future samples into either automatic or human evaluation. Note how the "pop" statement in the while loop always removes one element in the list, which results in a change in the error function. A key assumption here is human evaluators correctly classifies all samples they are provided.

**Listing 1.**    Python-like pseudocode for proposed uncertainty-sorting procedure

```
# X_val  - Sorted validation set
# U_val  - Sorted uncertainties
# f      - Classifier
# Er     - Error rate function
# e      - User specified error rate
# tau    - Uncertainty threshold

while Er(f, X_val) >= e:

    X_val.pop()

tau = U_val[len(X_val)]

return tau
```

For the process outlined in Listing 1 to be useful, it is critical that the uncertainty estimates $\sigma_i$ are informative and highlight samples where there is a high likelihood of error. There are numerous ways to estimate uncertainty, and below we describe three ways to estimate the uncertainties in $U^{(val)}$.

**Probability-based ranking**   While often criticized for being overconfident [6], the softmax probabilities of deep learning-based classification models can be used as confidence scores to filter out uncertain predictions, where scores closer to 0.5 are the most uncertain. Since the probabilities for oil being present or not can be confident in both ends (close to 1 or close to 0), we make the following transformation such that the confidence scores can be ranked:

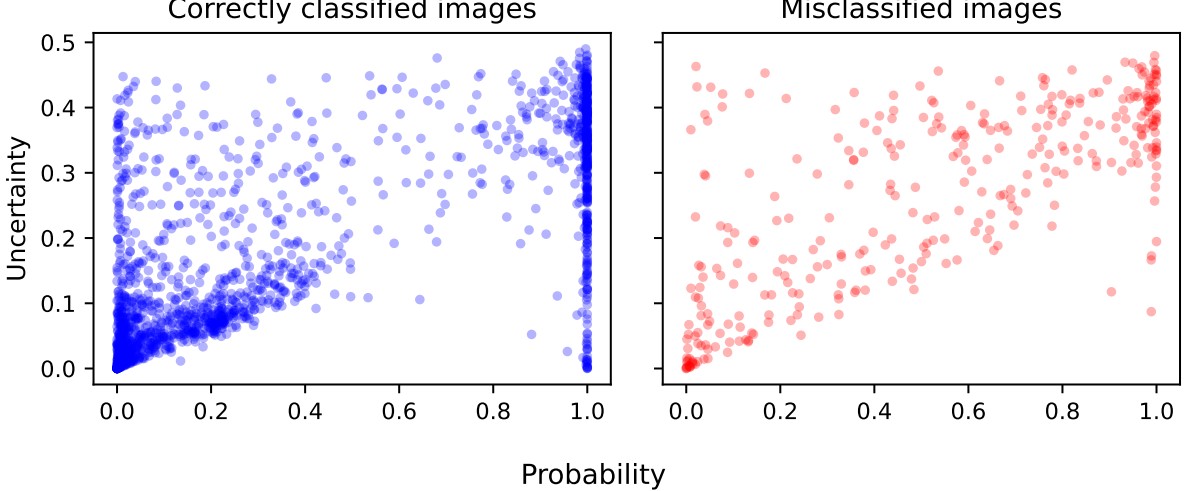

**Figure 3.** The standard deviation from the uncertainty estimation using PVS $(+/-, \gamma < 1)$ and the sigmoid values from the network for the testing set. Image on the left is for the correctly classified images and the misclassified images are on the right.

$$\sigma_i^{(p)} = |0.5 - \hat{y}_i|, \qquad (3)$$

where the $p$ indicates that the uncertainty comes from the output probabilities of the model.

**Uncertainty-based filtering** The standard deviation of the softmax output over the TTA samples described in Equation 1 can be used as an uncertainty estimation to identify uncertain predictions. The samples are ranked such that the images with the highest standard deviation are considered the most uncertain.

**Probability and uncertainty-based filtering** An alternative approach is to combine both the output probabilities and the uncertainty estimates to perform the filtering. Figure 3 shows the probabilities plotted versus the uncertainty estimates for correctly and incorrectly classified samples. A key observation here is that many misclassified samples have a probability of approximately 1, which means they will be among the last to be filtered out for human analysis. However, when taking uncertainty into account, many of the misclassified samples with an output of almost 1 also have high uncertainty, which means that they would be picked up in the filtering process. The complementary information in the probabilities and uncertainties could therefore provide complementary information for the ranking.

## 4 Experimental Setup

Here, we describe the data used in this study, the hyperparameters for each augmentation scheme, the model used to perform the oil spill detection, and how the model was trained.

**Data** The dataset consists of 313 SAR images taken from the Satellite Sentinel-1A. The images were preprocessed by downsampling to a $480 \times 480$ pixels corresponding to a resolution of 60m and cropped into 10317 patches of which 6960 patches fell into a chosen area of interest which were used for this study. The images were segmented into 7 classes: (i) Background: All pixels not explicitly as one of the following labels, present in 763 samples; (ii) Undefined: Low confidence oil spill, present in 1223 samples; (iii) Possible spill: Medium confidence oil spills, present in 497 samples; (iv) Probable spill: High confidence oil spills, present in 497 samples; (v) Seep: Oil seeping naturally from reservoirs at known seep locations, present in 65 samples; (vi) Produced water: Liquid byproduct of oil production, contain some oil as well as wastewater, present in 30 samples; (vii) Ignore: Boundaries of all oil spills and also missing data. These segmentation labels were compiled into binary image-level labels based on the occurrence of any of the oil spill classes (ii)-(vi).

The dataset is divided into 56% images for training (3926 images), 30% for testing (2050 images), and 14% for validation (984 images). The division is provided by Kongsberg Satellite Services (KSAT) and based on the acquisition days to ensure no data leakage between the splits as some images overlap and there are potentially several images over the same area each day.

**Data augmentation hyperparameters** For dropout, we investigate a range of hyperparameter ranging from a low dropout rate (0.01) to a

**Table 1.** Percentage of testing dataset needed to be manually analyzed to get a human error rate of 5%. Bold numbers indicate improved performance compared to the regular model. Results represent the mean and standard deviation across 3 independent training runs.

| Method | Sigmoid | Uncertainty (std) | Sigmoid + Uncertainty (std) |
|---|---|---|---|
| Regular model | 51 ± 10 % | - | - |
| PVS (+/-, $\gamma < 1$) | **42 ± 1 %** | **45 ± 4 %** | **42 ± 1 %** |
| Dropout (p=0.01) | 54 ± 11 % | **50 ± 9 %** | 54 ± 11 % |
| Elastic transform | **44 ± 4 %** | **41 ± 2 %** | **45 ± 4 %** |

high dropout rate (0.5). For PVS, we consider sampling $\gamma$ in Equation 2 from either $U(-a, b)$, $U(0, b)$, $U(-a, 0)$, with $a = b = 1$ or $a = b = 2$. Due to the computational load of the elastic transformation, we qualitatively identified a set of hyperparameters that induced some distortion and used those hyperparameters throughout all experiments. We use a grid size of $20 \times 20$ and a distortion magnitude of 5.

**Model and training** A ResNet-50 model [20] was trained for binary classification for 50 epochs using the Adam optimizer [21], binary cross-entropy loss, learning rate of 0.00001, weight decay of 0.0001 and batch size 16. As a stopping criteria, the highest AUC (area-under-the-curve) for the validation set is used. The network is initialized with pretrained weights using contrastive learning for Sentinel-1 and Sentinel-2 data [22, 23]. The weights are obtained from TorchGeo [24].

## 5 Results

Here, we present the evaluation of the proposed uncertainty-filtering approach to reduce manual workload in classification of oil spills in SAR images. We first evaluate the performance, before we present the quantitative evaluation for the proposed uncertainty-filtering procedure. Afterwards, we present an investigation of the effect of oil spill type and size on the proecdure before an in-depth analysis of the hyperparameters used in TTA. In Appendix A, we also investigate the added computational complexity of performing TTA.

**Model performance** A ResNet-50 is trained and classification performance is evaluated based on the experimental setup described in Section 4. The performance of the model was evaluated w.r.t. accuracy, F1 score, and AUC score. On the independent test set, the model achieves an accuracy of 83.6%, an AUC of 87.8%, and an F1 score of 75.9%. This model forms the basis for the following experiments in this section.

**Uncertainty-filtering reduces workload** We evaluate the probability-based, uncertainty-based, and the combined approach for uncertainty filtering with dropout, PVS, and elastic transform as the data augmentation. Table 1 shows the results from the best performing setup across all hyperparameters settings (see Table 2 for evaluation of hyperparameters), with the acceptable error rate set to 5 %. The results are from 3 independent training runs. Test for statistical significance was conducted but not found due to the low number of runs. Future works should repeat experiments across more runs to improve the rigor of the analysis. First, note that using the sigmoid output for the probability-based filtering ("regular model"-row) can already provide a reduction in the workload of the human evaluator. However, the greatest reduction occurse when uncertainty is taken into account through TTA, with a reduction of up to $\approx 40\%$. Concretely, the test dataset in this manuscript has 2050 images. Using the probability-based filtering reduces the amount of images need to be manually analyzed from 2050 to $\approx 1000$ images, and taking uncertainty into account reduces the number further down to $\approx 850$. This is a significant reduction in workload with great potential for practical gains. Interestingly, using TTA with dropout rarely leads to noteworthy improvements, which we attribute to its poor fit with the oil spill detection task.

Figure 4 shows the error on as a function of the percentage of samples sent to the operator. The plots show how TTA with PVS and elastic transformations requires less samples to be analyzed by the operator, which corroborates Table 1. In Appendix B, we also investigate the performance for a 1% error rate.

**Alignment between human and machine uncertainty of oil spill identification** The labels used for classification are either 0 (no oil) or 1 (oil), but more fine-grained labels in form of segmentation masks are available, as explained in Section 4. Figure 5 shows the uncertainty of images containing pixels of the respective segmentation classes. The boxplots are calculated using all images in the testing dataset that contain the specified class. The background class is in all images.

The uncertainty profiles of TTA based on PVS and elastic transformations differ notably: PVS yields

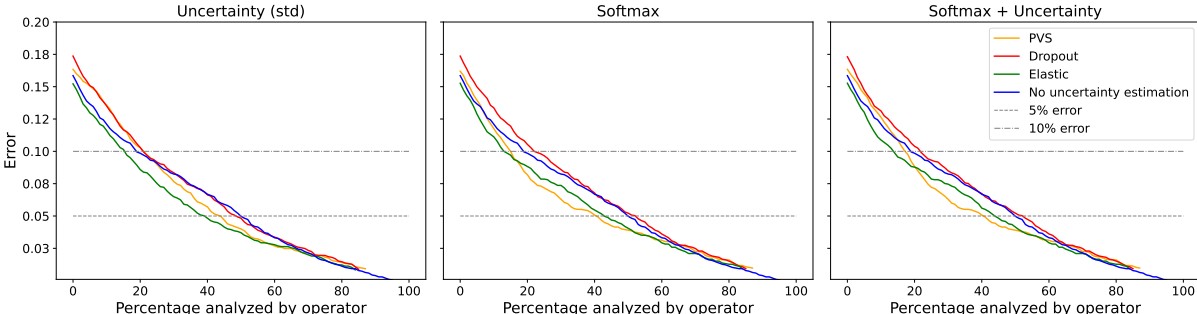

**Figure 4.** Plots showing the classification error with respect to the percentage of images that have to be analyzed by an operator. The three error thresholds are in grey. The three curves are from the methods with best performance, which are dropout with probability 0.01, PVS $(+/-, \gamma < 1)$ and elastic transformation. The first plot is the performance when sending the images with highest standard deviation. The second for sending the ones with most uncertain sigmoid values, which is the mean from the predictions using TTA. The last plot combines both methods.

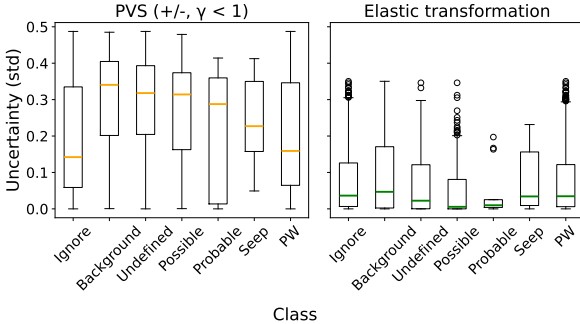

**Figure 5.** Boxplot of the standard deviation for the different classes in the pixel annotations. Plots are made using PVS $(+/-, \gamma < 1)$ as augmentation (left) and elastic transformation (right) as data augmentation.

generally high uncertainty (up to 0.5) with few outliers, while elastic transformations result in mostly low uncertainty (maximum 0.35) but with many outliers. This contrast likely arises from the nature of each augmentation. PVS uniformly shifts pixel values, altering image intensity and making samples less familiar to the model, whereas elastic transformation introduces local deformations but preserves overall intensity, resulting in lower uncertainty. Both methods show similar uncertainty for ignore and background classes, with PVS having slightly higher uncertainty for images containing the ignore class, likely due to its proximity to oil spills. Operator uncertainty for oil spill categories (undefined, possible, probable) aligns with average uncertainty values, and the seep class consistently shows the lowest uncertainty. Produced water exhibits the lowest uncertainty for PVS but nearly the highest for elastic transformation, though its limited representation makes conclusions difficult.

**On the effect of oil spill size** Another important factor that characterizes an oil spill is the size, which refers to the surface area the oil spill covers. Figure 6 shows the standard deviation from using PVS $(+/-, \gamma < 1)$ and elastic transformation as augmentation for the different oil spill sizes in the testing dataset. Each data point in the plot corresponds to the uncertainty and oil spill size for a testing image containing oil. It shows that images with small oil spills generally have high uncertainty for using PVS $(+/-, \gamma < 1)$. However, it is opposite for the elastic transformation. For this method, most images with small oil spills have low uncertainty.

Images with small oil spills tend to show high uncertainty when using the PVS $(+/-, \gamma < 1)$ method, likely because such spills are difficult to distinguish from natural features or may be partially visible near image borders, making classification challenging. In contrast, small oil spills generally exhibit low uncertainty with the elastic transformation method, possibly because most images have low uncertainty for this approach and the grid size used in the transformation means small spills may remain largely unaffected.

**Investigating TTA hyperparameters** A critical component of TTA is applying a suitable strength of augmentation, for example the amount of shift in PVS. In Table 2, we evaluate hyperparameters associated with PVS and Dropout. Due to the computational demand, we do not investigate the hyperparameters associated with the elastic transformation, as described in Section 4. For dropout, we see that the minimal amount of dropout noise gives the best performance, while only slightly increasing the dropout rate leads to much worse performance. We attribute this to the previously discussed hypothesis that dropout is not a suitable augmentation for oil spill detection due to its potential similarity with oil spills. For PVS, it is evident that pixel values should be shifted both in the positive and negative direction. Also, a weaker shift seems to be beneficial.

**Table 2.** Percentage of testing dataset needed to be manually analyzed to get a human error rate of 5% across different hyperparameters for each augmentation technique. Results represent the mean and standard deviation across 3 independent training runs.

| Method | Sigmoid | Uncertainty (std) | Sigmoid + Uncertainty |
|---|---|---|---|
| Regular model | 51 ± 10 % | - | - |
| PVS $(+/-, \gamma < 1)$ | **42 ± 1 %** | **45 ± 4 %** | **42 ± 1 %** |
| $PVS(+, \gamma < 1)$ | 60 ± 7 % | 58 ± 5 % | 60 ± 7 % |
| PVS $(-, \gamma < 1)$ | 54 ± 4 % | **46 ± 6 %** | 54 ± 4 % |
| PVS $(+/-, \gamma < 2)$ | **49 ± 7 %** | 52 ± 4 % | **49 ± 7 %** |
| PVS $(+, \gamma < 2)$ | 62 ± 5 % | 56 ± 5 % | 62 ± 5 % |
| PVS $(-, \gamma < 2)$ | 59 ± 6 % | 52 ± 7 % | 59 ± 6 % |
| Dropout (p=0.01) | 54 ± 11 % | **50 ± 9 %** | 54 ± 11 % |
| Dropout (p=0.1) | 70 ± 4 % | 66 ± 5 % | 70 ± 4 % |
| Dropout (p=0.25) | 79 ± 4 % | 77 ± 4 % | 79 ± 4 % |
| Dropout (p=0.5) | 84 ± 1 % | 82 ± 3 % | 84 ± 1 % |

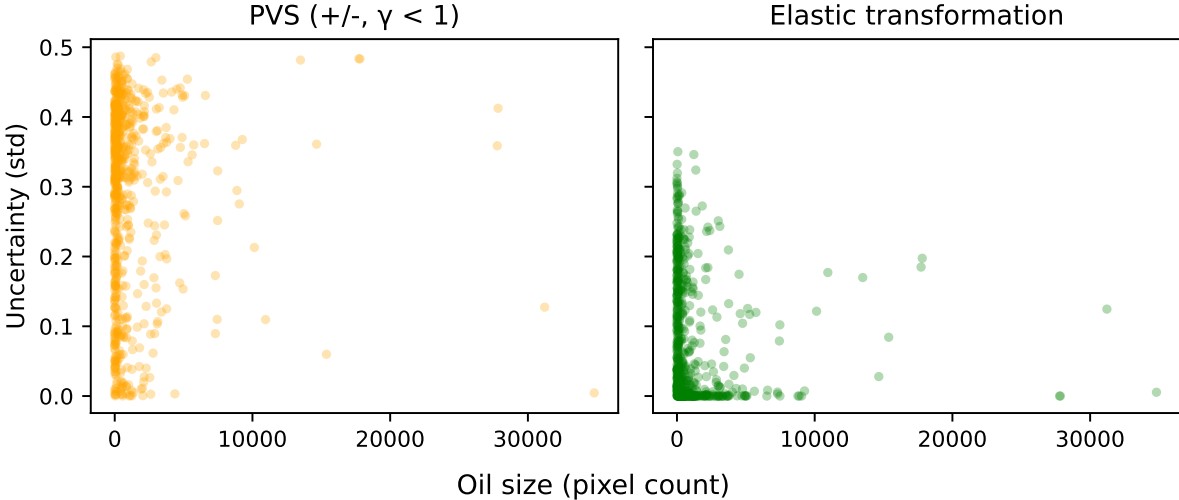

**Figure 6.** Plot of the standard deviation for each of the oil spill sizes in the testing dataset. Each circle represents one sample. Plots are made using PVS $(+/-, \gamma < 1)$ (left) and elastic transformation (right) as data augmentation.

## 6 Discussion

**On the compositions of augmentations** In our main results, we have only considered applying one type of augmentation in the TTA procedure. However, it has been shown that combing augmentations can lead to improved performance [17]. In Appendix C, we show some initial results where PVS and elastic transformations are combined, with encouraging results compared to the results in Table 1. However, the combination of augmentations introduces an additional element of complexity, as both the strength of the augmentations and the order in which they applied mush be investigated. We believe that future works could more thoroughly investigate this aspect of TTA in the context of uncertainty filtering.

**On the calibration of the classifier** We trained our model following standard procedures in deep learning (see Section 4). This shows encouraging results, but it is well known that the output probabilities might not be well-calibrated [6]. This certainly affects the output probabilities (see Figure 3), but could potentially also affect the uncertainty estimates. We investigated standard methods for improving the calibration of the model (see Appendix D), but saw little difference between the models with different calibration. An interesting line of future research could be to incorporate more sophisticated robustness strategies [25], to see if more calibrated classifiers could improve the filtering further.

**On the use of alternative uncertainty estimation techniques** This work focused on TTA to estimate uncertainty, which was motivated from the perspective of not having to change existing models. However, other uncertainty estimation methods like Monte Carlo Dropout [14], SWAG [15], or ensembles [16] could provide alternative or complementary uncertainty estimates. We believe that investigat-

ing alternative uncertainty estimation methods and potential combinations of different methods is an important line of future research, but beyond the scope of this work.

**Defining the human error rate** The selection of the human error rate is highly dependent on the particular task in questions. The ideal scenario would be to have domain experts with knowledge about the expected human error rate that can be directly used in the procedure. In this case, we did not have access to such privileged information, and evaluated a set of general error rate to illustrate the potential benefit of our proposed procedure. Another import aspect is the assumption of perfect performance of the human operators. This assumption might not always be realistic, since errors will always occur. However, we do believe it is reasonable to assume particularly strong performance from the human operators in this context, since they will have extra time to analyze and image and due to the awareness that image sent to the human operators are particularly complex. An alternative approach to relax the assumption of perfect performance from the human operators, namely to assume that there is a probability of the human operator making a mistake, would be an interesting line of future research.

# 7 Conclusion

We proposed an uncertainty-guided approach to reduce manual labor in oil spill detection from SAR imagery of the ocean. Given a user specified acceptable error threshold and an independent validation set of data, we automatically tune an uncertainty threshold to achieve a desirable trade-off between performance and efficiency. Our extensive evaluation on challenging real-world data shows that our proposed filtering approach can significantly reduce the manual workload associated with SAR-based oil spill detection. In future works, we envision to include explainability techniques [26] to further enhance the manual analysis aspect of the pipeline. We believe that with further developments our proposed uncertainty-filtering also has potential outside this particular application, and that it can play an important part in automated deep learning systems for industrial applications in years to come.

# 8 Acknowledgments

Funded by the Research Council of Norway, Visual Intelligence grant no. 309439. We would also like to thank the reviewers who had several important comments that affected the final version of the manuscript, particularly the discussion on how to select the human error rate.

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

# A  TTA computational time

We investigate the increase in computational time due to the TTA procesure. For a single image, we measure the computation time to take 0.0097 seconds. With M=30, PVS increases the time to 2.5102 seconds and elastic transform increases the time further to 5.0044 seconds.

# B  Additional error rate threshold

Figure 4 shows the error on as a function of the percentage of samples sent to the operator, where a human error rate of 1% is included. In the 1%, the human operators end up having to inspect a much greater portion of the data. We hypothesize that the reason for this is that the final 1% of errors are due to very small spills that the model cannot identify. Future works should therefore incorporate methods that encourage higher sensitivity to such small spills.

# C  Combining augmentations

Using first PVS(+/-, $\gamma$ ¡ 1) and then elastic transformation as data augmentation, results in performance similar, or slightly better than the two methods alone. However, the combination increases the computational complexity, making the slight increase in performance a relatively small gain. To get even

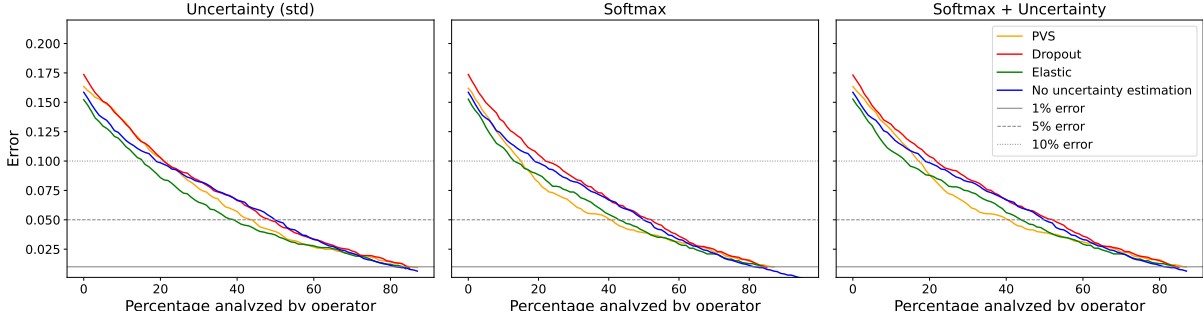

**Figure B.1.** Plots showing the classification error with respect to the percentage of images that have to be analyzed by an operator. The three error thresholds are in grey. The three curves are from the methods with best performance, which are dropout with probability 0.01, PVS $(+/-, \gamma < 1)$ and elastic transformation. The first plot is the performance when sending the images with highest standard deviation. The second for sending the ones with most uncertain sigmoid values, which is the mean from the predictions using TTA. The last plot combines both methods.

better performance using the combination, the parameters for the two methods should be tuned together. Using the combination in opposite order gives drastically worse results, seen in Table C.1. This might relate to the PVS being based on the data distribution of the training images. The elastic transformation shifts the values and possibly resulting in another distribution. The usage of PVS on the elastic transformed images might then not be appropriate, as the parameters estimated in PVS are not fitting anymore.

The composition of augmentations gives encouraging results, but a more rigorous analysis is necessary to conclude on the benefit. This analysis quickly becomes computational demanding, and we therefore consider such an analysis outside the scope of this work. There are several aspects that should be investigated. For instance, changing the order can completely change the performance. However, the strength of the augmentations must also be considered, as they the combination of augmentation can have an amplification effect. The positive results is for a set of hyperparameters, but notice that the spread in the performance increases. Initial experiments with other hyperparameters showed that this procedure could be quite unstable.

# D  Calibration

Temperature scaling is a standard way to prevent overconfidence by calibrating the network [6]. This is when the output from the network is divided by the temperature value of the network. The temperature is found using an optimization algorithm which minimizes the loss. The benefits of using temperature scaling on the model are explored. Results show that temperature scaling improves calibration, as shown in Table D.1. However, we found little improvement in terms of the uncertainty filtering, as seen in Figure D.1. We believe that this is because a single scaling factor is applied to the entire model, which might have less of an impact of the resulting ordering of samples.

**Table D.1.** The Negative Log Likelihood (NLL) and Expected Calibration Error (ECE) of the model using the validation set before and after temperature scaling. The temperature used for the network is 1.159

|  | Before temperature scaling | After temperature scaling |
|---|---|---|
| NLL | 0.573 | 0.574 |
| ECE | 0.466 | 0.439 |

**Table C.1.** Percentage of testing dataset needed to be manually analyzed to get a human error rate of 5% for the PVS $(+/-, \gamma < 1)$, elastic transformation and using both.

| Method | Sigmoid | Uncertainty (std) | Sigmoid + Uncertainty (std) |
|---|---|---|---|
| PVS $(+/-, \gamma <1)$ | $42 \pm 1$ % | $45 \pm 4$ % | $42 \pm 1$ % |
| Elastic transform | $44 \pm 4$ % | $41 \pm 2$ % | $45 \pm 4$ % |
| PVS + Elastic | $40 \pm 7$ % | $43 \pm 6$ % | $40 \pm 8$ % |
| Elastic + PVS | $77 \pm 4$ % | $79 \pm 6$ % | $77 \pm 4$ % |

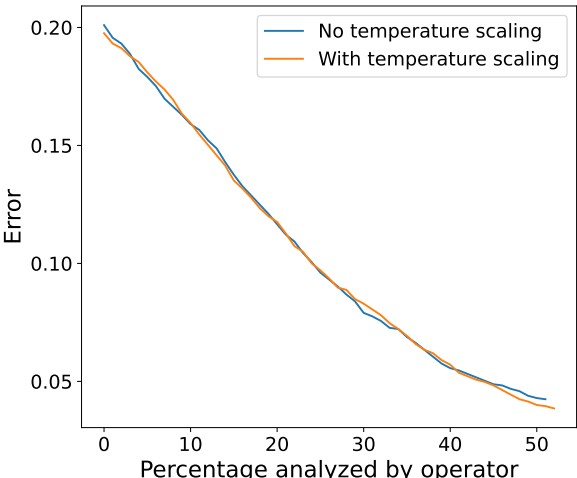

**Figure D.1.** The figure shows the error for the different percentages of images being sent to the operator, for both with and without temperature scaling in the ResNet-50 network. The images being sent to the operator are the ones with highest standard deviation using PVS (-, $\gamma < 1$)

