# OpenReview forum: "Reducing Manual Workload in SAR-Based Oil Spill Detection Through Uncertainty-Aware Deep Learning"
_NLDL.org/2026/Conference — NLDL 2026 Oral_

### Official Review · Reviewer_7UMx · 2025-10-07
**Review of using uncertainty-aware deep learning to reduce manual workload in the context of oil spill detection**

**Rating:** 4
**Confidence:** 3
**Final Rating:** 4
**Final Confidence:** 4

**Summary:**

Current deep learning (DL) models are not accurate enough to be the sole monitor of ocean imaging to look for oil spills. This work proposes a procedure that can quantify uncertainty to send the images that need more care to human operators for inspection while still reducing workload by letting all other images be classified automatically. This work does three main things. First, it proposes a novel procedure in DL classification for selection of the more uncertain samples to be sent to humans. Second, it identifies strategies for how to estimate uncertainty at inference time. Finally, it analyses a case study of trying these methods on real world data.

**Strengths:**

1. They clearly explain prior art for both the automatic detection models and ways of quantifying uncertainty. Then, they explain why they chose the methods they went with.
2. They clearly state assumptions such as “human evaluators will correctly classify all samples they are given”
3. They state limitations of the work like when they recognized that some of their data could have multiple images of the same area in the same day so they took that into consideration when splitting training and testing data
4. They have a reference method (called regular model) to compare performance against
5. They have various figures to show they studied different effects and different parameters to find the best method
6. They include confidence intervals on their data

**Weaknesses:**

1. If I understand correctly, for figure 3, the misclassified image plot should be most dense in the top left (high uncertainty and low probability of being correct). Why does the correctly classified plot have a higher density there?
2. Is there any information on what a reasonable error threshold (human error rate) is so the reader knows how reasonable the 5% used in this work was?
3. More detail could be provided to clarify some points about the error function mentioned in Section 3.2 and Listing 1. It seems to be used to measure the error rate for each sample using model accuracy. However, how does it get a different value for each sample if it only uses previously calculated model accuracies on new samples at inference time? Does it also take into account the uncertainty to estimate error?

Less significant weaknesses:
1. Could add more explanation about why they did not use explainable AI techniques that would utilize intermediate neuron values of DL models as opposed to just the softmax layer in calculations. Or add to future work.
2. Could state how many samples per circle in Figure 3 to give more clarity in the figures
3. Could have explored a procedure for if we assume the human operators have some probability of also not being able to classify the sample correctly. Unless it is difficult to find literature values for this.

**Final Justification:**

This work combines state of the art methods in a novel way and applies them to a high impact application. The work explains limitations and possible future work. The authors also addressed my concerns in their response and said they will add tweaks to the manuscript for more clarity. I believe this has novelty worth sharing.

**Justification:**

Most of the weaknesses are just things to clarify, some literature context to add, and some things to add regarding future work. This work has explanations for the methods it chose to use and combined them in a novel way to apply them to a non-trivial and important application. They also mention limitations and compare their methods to a reference method to confirm better performance.

---

### Official Review · Reviewer_xWz6 · 2025-10-08
**Good written paper that applies novel deep learning techniques to SAR-based oil spill detection**

**Rating:** 4
**Confidence:** 3
**Final Rating:** 4
**Final Confidence:** 4

**Summary:**

The paper outlines a method how a hybrid system (automatic classification + manual analysis) can be used to reduce workload efficiently while remaining within a user defined error tolerance. They propose to use uncertainty estimation to effectively determine which samples need manual analysis. How the uncertainty can be estimated for SAR images is revealed in the paper.

**Strengths:**

It applies an interesting idea, uncertainty estimation with test-time augmentation, to the field of SAR based oil spill detection.
Domain specific augmentations are proposed with a good scientific analysis.

Looking at it directly from the point of view from the process used in the field is also refreshing, as this shows how deep learning can be used to make an impact directly.

**Weaknesses:**

The novelty could be emphasised more in the abstract, it is mentioned in the introduction but that is already on page 2.

The literature study could be elaborated, e.g. are there other fields where such a method has been applied successfully?

The composition of augmentations is only mentioned in the appendix, but since it improves the results why not mention it directly in the results in the paper?

**Final Justification:**

I would be inclined to accept it since a good effort was put in the rebuttal where most of my questions were answered.

**Justification:**

The paper explores successfully how uncertainty estimation can be used to speed up SAR-based oil spill detection. It further provides a good scientific analysis of the proposed method.

---

### Official Review · Reviewer_h3T6 · 2025-10-08
**A good application of deep learning methods for oil spill detection, image classification**

**Rating:** 4
**Confidence:** 4

**Summary:**

The paper addresses the challenge of reducing manual workload in SAR-based oil spill detection to reduce environmental damage. The authors propose an uncertainty-guided filtering approach that automatically identifies which images require human review versus automatic classification. The core technical innovation is using Test-Time Augmentation (TTA) to estimate prediction uncertainty, where multiple augmented versions of each input image are generated and the standard deviation across predictions serves as an uncertainty measure.

The authors investigate three augmentation strategies: Pixel Value Shift (PVS), elastic transformations, and dropout. They propose an automated threshold selection algorithm that, given a user-specified acceptable error rate and a validation set, determines an uncertainty threshold to optimally split samples between automatic and manual processing. The method is evaluated on 6,960 SAR image patches from Sentinel-1A satellite data using a ResNet-50 classifier. Results demonstrate that using domain-specific augmentations (particularly PVS with bidirectional shifts) reduces manual workload by approximately 41% while maintaining a 5% error threshold, decreasing the number of images requiring manual review from 2,050 to 850.

The key insight is that combining probability-based confidence scores with TTA-derived uncertainty estimates identifies misclassified samples more effectively than either approach alone, as some high-confidence predictions also exhibit high uncertainty and would otherwise be missed by probability-based filtering alone.

**Strengths:**

**1. Novel and Practical Contribution**

The uncertainty-guided filtering approach addresses a real industrial problem with measurable impact. The 41% workload reduction represents significant practical value for operational oil spill monitoring, making it feasible in real-world settings.

**2. Domain-Specific Augmentation Strategy**

The authors provide strong intuition and empirical evidence for why domain-specific augmentations (PVS and elastic transformations) significantly outperform generic approaches like dropout for SAR imagery. The explanation that dropping dark pixels (representing oil spills) doesn't meaningfully alter appearance is well-reasoned, and the results in Table 1 clearly validate this hypothesis (dropout shows minimal improvement at 50-54% vs. 42% for PVS).

**3. Thorough Experimental Design**

The paper includes appropriate baselines, comprehensive hyperparameter studies (Table 2), and multiple ranking strategies (probability-based, uncertainty-based, and combined). The data split by acquisition date to prevent temporal leakage is methodologically sound.

**4. Clear Presentation and Visualization**

Figure 3's scatter plots effectively illustrate why combining probability and uncertainty is necessary, showing that many misclassified samples have high confidence (1.0) but also high uncertainty. Figure 4 clearly demonstrates the performance gains across different methods. The paper is generally well-written and accessible.

**5. Honest Discussion of Limitations**

The authors acknowledge important limitations in Section 6 (Discussion), including the need for exploring augmentation combinations and calibration techniques, and suggest future research directions in Appendices A and B.

**Weaknesses:**

**1. Critical Unrealistic Assumption**

The entire framework assumes perfect human classification (stated in Section 3.2, line 213-214). This is unrealistic—human operators make errors, especially on ambiguous cases. The paper should evaluate how robust the method is to human error rates (e.g., 1-3% human error) and discuss how this affects the overall system performance. The 41% workload reduction may be overstated if human errors are considered.

**2. Limited Comparison to Other Uncertainty Estimation Methods**

While the paper mentions Monte Carlo Dropout and ensemble methods in Section 3.1, these are never empirically compared. The claim that TTA is chosen for "flexibility" in line 119 because it requires no model modifications is practical, but the paper doesn't validate whether TTA produces better uncertainty estimates than these alternatives for this specific application. A comparison with at least one other method (e.g., MC Dropout with the same ResNet-50) would strengthen the contribution.

**3. Single Error Threshold Evaluation**

The paper only thoroughly evaluates performance at a 5% error threshold. Different operational contexts may require different risk tolerances (e.g., 1% for environmentally sensitive areas, 10% for lower-risk regions). Figure 4 shows some results at 10% error, but there's no detailed analysis. How does the workload reduction scale with different error tolerances? This is crucial for assessing practical applicability.

How does performance scale across different error thresholds (1%, 2%, 5%, 10%)?

Is a 41% manual workload reduction consistent across these different error thresholds?

**4. Insufficient Analysis of Computational Cost**

The paper mentions that elastic transformations are computationally expensive (Section 4, line 311-312), limiting hyperparameter exploration, but provides no concrete timing analysis. For real-time operational deployment, how much overhead does TTA add? If PVS requires generating M augmented versions per image, what is the latency impact? This is essential information for practitioners.

**5. Limited Scope and Generalizability Claims**

The conclusion states the approach "has potential outside this particular application" (line 487-488), but this is not demonstrated. The paper evaluates only one dataset, one domain (oil spills), and one architecture (ResNet-50). While the filtering method should be architecture-agnostic, demonstrating this on at least one additional model would strengthen generalizability claims. The dropout hypothesis is specific to this visual task; would it generalize to other SAR applications?

**6. Statistical Rigor**

The results in Tables 1 and 2 report standard deviations but no significance testing. Is the difference between PVS (42±1%) and elastic transforms (41±2%) statistically significant? The high variance in the baseline method (51±10%) versus PVS (42±1%) is notable in tabels 1 and 2 but not explained—is this due to random initialization, data splits, or something else?

**7. Incomplete Ablation on Augmentation Combinations**

Appendix A mentions that combining PVS and elastic transformations shows "performance similar, or slightly better than the two methods
alone" but this is not thoroughly explored in the main paper. Given that both augmentations capture different aspects (intensity vs. geometry), this seems like an important avenue that deserves more than a brief appendix mention.

**Justification:**

The research addresses a real problem with a practical solution and demonstrates meaningful results. The uncertainty-based filtering approach has novelty and the 41% workload reduction while maintaining safety standards is valuable for operational deployment. However, the unrealistic human performance assumption, limited comparison to alternative uncertainty methods, and questions about generalizability are present. With revisions addressing the computational cost analysis, robustness to human error, and comparison to at least one alternative uncertainty estimation method, this could be a strong contribution to the field.

This paper makes a solid contribution to practical oil spill detection systems with a novel uncertainty-guided filtering approach that achieves meaningful workload reduction. The domain-specific augmentation strategy is well-motivated and the experimental evaluation is generally thorough. However, the work would be significantly strengthened by: (1) relaxing the perfect human assumption, (2) comparing with other uncertainty methods, (3) analysing computational costs, and (4) demonstrating broader applicability.

---

### Official Review · Reviewer_KmQD · 2025-10-08
**Using Deep Learning and Uncertainty to Reduce Human Manual Verification Workload in Oil Spill Detection from SAR Images**

**Rating:** 4
**Confidence:** 4
**Final Rating:** 4
**Final Confidence:** 4

**Summary:**

This paper investigates whether uncertainty estimates can help identify which samples require manual verification in the task of oil spill detection from SAR imagery. The authors propose an uncertainty-guided selection process to determine which SAR images, automatically flagged as potential oil spills by a deep learning model, should be reviewed by a human evaluator.

An uncertainty threshold is automatically determined to decide when human verification is necessary. In addition to output-based uncertainty estimation, the approach employs test-time augmentation (TTA) to estimate uncertainty, which has the advantage of not requiring any modification to the underlying model.

**Strengths:**

+ The paper is well written and easy to read.
+ The evaluation is conducted on challenging real-world Sentinel-1 data.
+ The results show that using uncertainty-based selection for human evaluation can effectively reduce operators’ workload.
+ The proposed selection method is tested for both classification and segmentation tasks.
+ The authors provide an in-depth analysis of the results.

**Weaknesses:**

- No comparison with other methods from the literature was conducted.
- In Equation (1), a square term appears to be missing.
- It would be valuable if the authors included results obtained using Monte Carlo Dropout or SWAG for comparison.
- It is not clear how the human error rate was defined or estimated.

**Final Justification:**

I keep my rating as weak accept. The paper presents an interesting and practical application of uncertainty estimation for oil spill detection from SAR imagery. I think that the paragraph to be added to the discussion and the discussion about the selection of the error rate in the appendix will make the paper clearer.

**Justification:**

Despite the weaknesses, I consider this submission borderline. I lean toward accept, as the results demonstrate that the proposed uncertainty-based selection method effectively reduces human evaluation effort. Furthermore, although uncertainty estimation is widely used in active learning and the paper lacks comparisons with related methods, it presents an interesting and practical application of uncertainty estimation for oil spill detection from SAR imagery.

---

### Meta-Review · Area_Chair_kWDA · 2025-11-01

**Recommendation:** Accept (Oral)
**Confidence:** 4

**Metareview:**

Firstly, the authors engaged well by providing detailed responses to the reviewers' comments, which helped to clarify some issues.

Now, in terms of specifics, this paper presents a well-motivated, practically relevant approach to reducing manual workload in SAR-based oil spill detection by leveraging uncertainty-aware deep learning. The authors propose a pipeline that uses test-time augmentation (TTA) to estimate prediction uncertainty and automatically filter samples for manual review based on a user-defined error tolerance. The method is evaluated on real-world Sentinel-1 SAR data and achieves a 41% reduction in manual workload while maintaining an error threshold of 5%
.
The reviewers consistently praised the clarity of the writing, the novelty of the proposed uncertainty-guided filtering procedure, and the practical impact of the work. As I mentioned above, the authors provided detailed and constructive rebuttals that addressed all major concerns, including assumptions about human performance, computational cost, and generalisability.

Therefore, I think this paper would make a very good contribution to the conference.

---

### Decision · Program_Chairs · 2025-11-05

**Decision:**

Accept (Oral)

**Comment:**

We recommend an oral and a poster presentation given the AC and reviewers recommendations.